# Bridging the Gap between Clinical Service and Academic Education of Hand-Splinting Practice: Perspectives and Experiences of Thai Occupational Therapists

**DOI:** 10.3390/ijerph19158995

**Published:** 2022-07-24

**Authors:** Anuchart Kaunnil, Veerawat Sansri, Surachart Thongchoomsin, Kannika Permpoonputtana, Mandy Stanley, Piyawat Trevittaya, Chirathip Thawisuk, Peeradech Thichanpiang

**Affiliations:** 1Department of Occupational Therapy, Faculty of Associated Medical Sciences, Chiang Mai University, Chiang Mai 50200, Thailand; anuchart.kau@cmu.ac.th (A.K.); piya.trevit@cmu.ac.th (P.T.); 2Department of Basic Medical Science, Faculty of Medicine Vajira Hospital, Navamindradhiraj University, Bangkok 10300, Thailand; veerawat@nmu.ac.th; 3Division of Occupational Therapy, Faculty of Physical Therapy, Mahidol University, Nakhon Pathom 73170, Thailand; surachart.suk@mahidol.edu; 4National Institute for Child and Family Development, Mahidol University, Nakhon Pathom 73170, Thailand; kannika.per@mahidol.ac.th; 5School of Medical and Health Sciences, Edith Cowan University, Joondalup, WA 6027, Australia; m.stanley@ecu.edu.au; 6Department of Occupational Therapy, Graduate School of Human Health Sciences, Tokyo Metropolitan University, Tokyo 191-0065, Japan; thawisuk-chirathip@ed.tmu.ac.jp

**Keywords:** occupational therapy, splinting practice, orthosis

## Abstract

A gap in knowledge about current splinting practice exists between the educational program and clinical service. To bridge this gap, we investigated the perspectives and experiences of Thai occupational therapists regarding contemporary hand splinting practices in clinical use. A mixed-method study was designed. An explanatory sequential mixed methods design was used. In the first quantitative phase, a survey questionnaire was mailed to occupational therapists. The questions were regarding contemporary hand splinting practices in clinical use at seven hospitals in the capital city of Bangkok and outskirt areas. In the second phase, semi-structured interviews were completed to explore expert occupational therapists’ perspectives on practice in the same hospital settings. Transcripts were analyzed using thematic analysis. The results showed that most conditions receiving splints were nerve injuries, orthopedics, and stroke, which represented the service frequency of splint types: functional resting (100%), cock-up (93.3%), and thumb spica splints (80%). Bone and joint deformity prevention ranked first with muscle contracture prevention being ranked second, and the third-ranked was maintaining range of motion. Three themes emerged from the interviews: starting with the patient condition; effective function and value; knowledge and experiential skills. Perspectives and experiences of occupational therapists in splinting practice contribute to education based on the reality of practice. Integrated numerical and textual data of professional skills and knowledge in actual splinting practice can be reflected through splints and orthoses program revisions to meet future learning outcomes.

## 1. Introduction

Occupational therapy is founded on human activities in the dynamic processes of engagement in physical, socio-cultural, temporal, psychological, spiritual, and virtual environments [1]. Therefore, the education to become an occupational therapist has to incorporate all of those elements and teach clinical reasoning processes. Occupational therapy education should teach occupational therapy students to blend theoretical knowledge, skills, and experience to meet the learning outcomes. The learning results can be improved through the content and scope of the study program based on competence in knowledge, skills, and attitudes to generate capable, competent occupational therapy professionals [2,3].

In Thailand, The Ministry of Higher Education, Science, Research, and Innovation enacted laws and regulations for Thai universities. Occupational therapy programs require the Thai Qualifications Framework (TQF) for Higher Education be implemented [4], and the Professional Standards for the Art of Healing in Occupational Therapy [5]. Students take four years to earn their Bachelor of Science (Occupational Therapy) degree with an eight-semester curriculum. The Thai occupational therapy program is comprised of general education courses, specialized or professional courses, and free elective courses. Thailand meets the World Federation of Occupational Therapists (WFOT) requirement that students complete a minimum of 1000 h of fieldwork experience as part of the occupational therapy curriculum [6].

A major goal of undergraduate occupational therapy programs is to prepare competent and capable entry-level therapists [7]. Clinical reasoning skills are one of the most challenging aspects of the occupational therapy program [8]. As a general term, clinical reasoning can be described as the process of gathering and interpreting information in order to make decisions regarding health care [9]. Other terms used interchangeably with clinical reasoning are clinical judgment, decision-making process, and critical thinking reasoning [10]. In health professional education, clinical reasoning is considered a core component of teaching and assessment [11]. A progressive approach to clinical reasoning skills should be employed during the education program [12]. Occupational therapists should supervise final-year students during their fieldwork education to ensure learning takes place at a deep level of reasoning [13]. Hence, clinical reasoning is a primary skill in occupational therapy education for identifying, prioritizing, establishing plans, and interpreting clinical data, which are all important to occupational therapy practice.

The traditional educational curriculum for occupational therapy students relies on didactic lectures and laboratories for practicing skills, reinforced with one-on-one or group teaching and learning processes. The occupational therapy programs have been approved by the WFOT; however, the program, courses, and learning outcomes were adopted based on Western styles. Some courses, such as splinting and orthoses fabrication, do not consider the prevalent conditions and most frequent types of splints in use in the current services and contexts. This is a gap in the current knowledge in achieving learning outcomes and professional skills for occupational therapy students. In Japan, a fundamental aspect of occupational therapy involves incorporating societal needs and cultural considerations in the occupational therapy process [14], as cultural tensions affect clinical practice.

Currently, most Thai universities have embraced the outcome-based education (OBE) approach driven by the ASEAN University Network-Quality Assurance (AUN-QA) to promote students to meet learning outcomes [15]. To achieve OBE, the mechanisms and frameworks were established for promoting, developing, planning, organizing, monitoring, and evaluating educational programs and learning activities [16]. The occupational therapy programs of Thai universities have been implemented in the curriculum and courses through AUN-QA. A course coordinator of splints and orthoses reported competency deficiencies in the knowledge and core skills among occupational therapy students. This information was received from occupational therapy clinical instructors of students’ internships and placements in Bangkok and the surrounding areas. This motivated us to find current knowledge and experiential skills related to splinting practice during service.

The primary goals of providing splint services for people are to reduce pain, slow the degradation process, recover joint space, and improve finger performance and hand functioning by increasing both grip and pinch strength. Such splinting practice enables patients to perform everyday activities and was shown to enhance their quality of life [14]. In the U.K., Kilbride et al. [17] studied contemporary splinting practices of British physical therapists (*n* = 159) and occupational therapists (*n* = 261) for adults with neurological disorders. They examined current splinting practices in contracture management and noted the similarities and differences in the splinting practices of occupational therapists and physiotherapists. These findings helped in constructing a national guideline for splinting in neurological rehabilitation.

In occupational therapy programs, the course of splints and orthoses is an integral aspect of students’ knowledge and skills, which include the types of splints and conditions based on population context, as splinting practice is a preparatory approach to maintaining joint position in both static and dynamic situations. These therapeutic means are key to improving abnormal muscle tone and increasing range of motion and can help overcome soft tissue and joint contractures resulting from injuries [18]. Such splinting practice enables people to perform everyday occupations and has been shown to enhance the quality of life [19].

In Australia, occupational therapists and physiotherapists were surveyed about hand therapy services in rural and remote areas. Most therapists provided initial splinting and exercise practice. They reported experiencing many barriers in providing service in rural/ remote areas in terms of transport, time, staffing, and providing specialist service for patients with hand injuries. This situation may be solved by professional support via videoconferencing and technology to improve equity and resolve the patients’ problems [20]. In the United States, a survey was conducted to explore the understanding of occupational therapists in 21 different states on contemporary splinting practices for patients with cervical spinal cord injury. The study found that the main principle for splinting and service was hand function. Resting hand splints without active arm movement were used for patients at night, and wrist splints without active arm movement were used for daytime [21].

In Thailand, only two studies have been published on splinting. Kumnerddee and Kaewtong [22] studied the efficacy of volar neutral wrist splints at night compared to acupuncture for carpal tunnel syndrome over five weeks. They found no significant difference between the groups in terms of symptom severity or hand function; however, pain, as measured on a visual analog scale (VAS), was significantly less for the patients who received acupuncture than for those with night splints (*p* = 0.028). In 2016, Bunyachatakul et al. [23] studied the effects of a low-profile traction dynamic hand splint system using low-cost materials for patients with radial nerve injury. Hand function was measured by the Modified Southampton Hand Assessment Procedure (MSHAP), and semi-structured interviews were used to evaluate splint users’ level of satisfaction. Use of the low-profile dynamic hand splint did not show an improvement in scores of hand function compared to the baseline. Participants were satisfied with the biomechanical efficacy of these splints, the functional performance of the splint in daily life, and the ease of splint maintenance.

### Purpose of the Study

However, there are no studies regarding splinting practices related to university education programs based on the evidence of real practice. Splinting practice in Thailand may be different from in Europe and North America due to the prevalence and differences in the conditions of patient populations and socio-cultural contexts. The purpose of this mixed methods sequential explanatory study was to identify occupational therapists’ perspectives in splinting practice. The specific objectives were:○To explore occupational therapists’ perspectives on clients’ conditions and factors, splint types, and rationale for splinting selection and practice.○To find the experience of expert occupational therapists in the nature of clinical instructors determining their clinical reasoning ability and skill in splinting practice.

The outcomes of the two specific objectives will help shape the scope and content of courses on splinting and orthotics in occupational therapy program revisions.

## 2. Materials and Methods

This study addressed the perspectives and experiences of occupational therapists on upper limb and hand splinting practices. The explanatory sequential mixed methods design [24] involved collecting quantitative data first and then explaining the quantitative results with in-depth qualitative data. Ethical approval for this study was obtained from the Mahidol University Central Institutional Review Board, protocol number MU-CIRB 2018/142.1007 (COA No. MU-CIRB 2018/144.1508). Written informed consent was obtained from legally authorized representatives before the study.

### 2.1. Survey Procedure

A questionnaire survey was developed by the research team and contained questions in five parts: (1) demographic profile, workplace location, and occupational therapy specialty; (2) experience in splinting production; (3) types of splinting designs used in practice and fabrication; (4) patient factors for splinting practice; (5) therapist factors in splinting practice. Prior to administering the survey, Item Objective Congruence (IOC) was used to evaluate the items of the questionnaire based on the scoring system from −1 to +1 (congruent/agree = +1, questionable = 0, and incongruent/disagree = −1). The questionnaire items with scores higher than or equal to 0.5 were retained. Conversely, the items with scores lower than 0.5 were revised [25].

The questionnaire was validated by three external occupational therapists who had more than 20 years experience with splinting. This questionnaire received a score of 0.73 on the Item Objective Congruence (IOC) index.

The questions asked in the questionnaire were: Which diseases and injuries are commonly seen in your splinting practice? What splints are used most frequently in service? Participants were asked about the frequency of patients’ conditions at the splinting service and the types of splints (use or never use). Participants were asked to rank their opinion using a five-point Likert scale—(1) very frequently, (2) frequently, (3) occasionally, (4) rarely, and (5) never—about patient factors in splinting practice.

Lastly, respondents were asked about their rationale for splinting selection and application. This questionnaire required approximately 15–20 min to complete.

### 2.2. Survey Participants and Recruitment

There are seven tertiary hospitals in the Bangkok metropolitan area where occupational therapists provide hand therapy in rehabilitation settings and work as clinical occupational therapy instructors. Forty-two occupational therapy practitioners were selected by purposeful sampling based on the following criteria: (1) possessing a national occupational therapy license; (2) having worked as clinical instructors (3) having more than two years of experience in splinting service; and (4) working full-time. In Thailand, clinical instructors are mainly employed by the government, providing practical knowledge in a particular clinical area. In splinting practice, they teach occupational therapy students splint fabrication and provide splints for patients during a fieldwork placement. Advertisements and contact forms were sent to seven hospitals to invite occupational therapists based on the inclusion criteria to join the survey study.

### 2.3. Interview Procedure

The interview guidelines were designed and refined by an iterative process involving mock individual in-depth interviews prior to its use. The semi-structured interviews comprised questions about (1) patient factors for splint fabrication, (2) development and function of splinting practice, and (3) therapist factors affecting splinting. The interview process was organized in a natural and logical sequence and conducted in a location offering a comfortable setting and relaxed atmosphere. Semi-structured interviews comprised:

Conditions related to splints: (i) What are the patient factors that impact your decision to provide splints?

Types of splints and function: (ii) What are your expectations for splint function and application?

Therapist factors: (iii) What is your rationale for choosing splint fitting with patients? (iv) How do you find evidence for selecting and providing splints for patients? (v) What are the factors that affect your confidence in providing splints for patients?

### 2.4. Interview Participants and Recruitment

Participants were recruited based on the following criteria: (1) possessing a national occupational therapy license; (2) having more than 10 years of experience and identifying as an expert in splinting service; (3) working as a clinical instructor; (4) working in the city of Bangkok and surrounding areas; (5) willing to participate in an interview. The individual interview participants were excluded from the survey to avoid data repetition. Advertisements were sent to hospitals and health institutes in Bangkok and the surrounding areas over three months. After participants were contacted by the research team, they were selected based on the inclusion criteria.

### 2.5. Data Collection Procedure

Questionnaires and consent forms with cover letters and stamped return envelopes were sent to 42 participants at seven hospitals based on the research site criteria in September and October 2018. All participants read and signed written consent forms before answering the questionnaire. Information was gathered on the nature of splinting practices and services in the participants’ hand rehabilitation settings, but the names of organizations and addresses were not identified.

Interview data were collected in February and March 2019. The researcher contacted the occupational therapists based on the inclusion criteria to arrange the most convenient time and location for an individual interview. This was conducted by an author following a semi-structured interview guide. Each participant was required to read and sign a written consent form before the interview. This interview was audio-recorded and transcribed verbatim, and the in-depth interview required approximately 1.5 h. All research processes were clearly documented, and the transcripts were peer-reviewed by the team. Pseudonyms were used to protect the identity of the participants.

### 2.6. Data Analysis

The survey data were analyzed using descriptive statistics and frequencies. All research processes were clearly documented, and the transcripts were checked by the research team. In-depth interview data were analyzed by thematic analysis [26]. This analytical process involved six steps. First, an overview of the data was performed to gain a general understanding of the occupational therapists’ experience. This process involved transcribing the audio verbatim onto paper and then typing it into Microsoft Word, reading through the text several times, and taking notes to become familiar with the data. Second, initial codes were generated for the data by highlighting sections of the text—phrases or sentences—to describe their content. These codes provide an overview of the main points and groups of common meanings that recur throughout the data. Third, patterns were identified among the codes and grouped into potential themes. Fourth, the themes were reviewed to ensure they were useful and accurate representations of the data. Fifth, the themes were defined and named to formulate the results. Sixth, the report was produced with reference to the research questions and the extant literature.

### 2.7. Rigor and Trustworthiness

This study examined potential translation-related problems which might interfere with trustworthiness. Word-for-word transcripts were analyzed in the native language (Thai) and then translated into English [27] and checked by two academics fluent in both Thai and English. In order to maintain semantic equivalence with realistic and textual meanings, linguistic differences were considered, for example, an expression that may exist in one language may not be present in another [28]. An audit trail was created by our research team to provide support for interpretations and analyses [29]. In addition, an adequate translation contributes to the trustworthiness of the study, since it contributes to accurate data, inclusivity, consistency, and transparency during the analytical process [26]. As part of increasing data transferability, the research team documented the process and peer-reviewed the transcripts. Three independent reviewers spent a significant amount of time on data translation, illustrating the rigor of the research. Moreover, participants were invited to validate the findings by confirming or suggesting variant meanings that could then be incorporated into the final statement.

## 3. Results

### 3.1. Survey Outcomes

The questionnaires were returned by the end of December 2018. There were 30 (71%) replies from 6 (20%) men and 24 (80%) women from seven settings with a mean age of 36.8 years (SD = 10). The participants had between 2 and more than 20 years of experience (mean ± standard deviation, 10.5 ± 7 years) in splinting practice within occupational therapy (Table 1).

Four male and four female occupational therapy specialists participated in the interviews. Six occupational therapists worked at public hospitals and two occupational therapists worked at private hospitals (Table 2). The mean age of respondents willing to participate in individual interviews was 43.8 years (SD = 4 years). They had between 15 and 25 years of experience (mean ± standard deviation = 2 years 0 ± 4) in splinting practice within occupational therapy.

Figure 1 shows the percentage distribution of patient conditions in splinting. Nerve injuries were ranked first (31%) among the participants that provided splint therapy. Orthopedic conditions were ranked second (24%), followed by stroke and traumatic brain injury ranked third (17%), with the remaining conditions showing low percentages (2%) of use among the participants providing splinting services.

Table 3 shows the percentage distribution of the most frequently used splints. The functional resting splint was ranked first with 100% usage within splinting services. Cock-up splints were ranked second with 93.3% usage, and both finger and thumb spica splints ranked third with 80% usage within the participants’ splinting services. Supination and weight-bearing splints were only used in 3.3% of splinting services. This analysis had participants rate their answers using a 5-point Likert scale: (1) never, (2) rarely, (3) occasionally, (4) frequently, and (5) very frequently. Participants indicated their clinical rationales and decision-making when selecting splints for their patients (Table 4).

A high percentage of participants reported that splinting practice is used to prevent bone and joint deformities (43% very frequently and 53% frequently) and could be used to prevent muscle contractures (40% very frequently and 46% frequently). Most participants provided splinting for maintaining and increasing range of motion (33% very frequently and 53% frequently). Moreover, participants used splinting practice to reduce muscle tone (27% very frequently and 50% frequently). The majority also believed that splinting interventions could improve hand function and prehension (10% very frequently, 47% frequently, and 20% occasionally). However, only 7% reported very frequently and 13% frequently using splinting for patients’ needs, with a low percentage of splinting used for pain relief (6% very frequently and 13% frequently).

Figure 2 shows the percentage distribution of the reasoning for splint selection and application based on the participants’ considerations. Clinical knowledge of the condition was ranked first with 35% of the participants when selecting splinting for patients. Splinting skills and experience were ranked second (30%), and the availability and accessibility of splinting were ranked third (15%). Both raw materials and design of splint ranked fourth (8%), with the patients’ needs showing only 4% in the participants’ rationale for splint selection.

### 3.2. Qualitative Findings

Three themes arose from the analysis of the interview data in Figure 3: (1) starting with the patient condition; (2) effective function and patients’ value; (3) knowledge and skills. These themes were integrated with the survey findings, in which textual and numerical results provided outcomes related to splinting practice.


**Theme 1. Starting with the patients’ condition**


The conditions and types of splints are fundamental principles that need to be considered in splinting practice. Splint selection based on the patients’ condition needs to be the first point of consideration, which in turn can lead to improved hand function and performance during functional tasks. Participants knew the causes, conditions, and clinical implications of the upper limbs or hands of patients. In Somchai’s account, they reflected on the treatment process once the condition was diagnosed by a physician and consultation with an occupational therapist had taken place to create a splint for a patient, as stated in the following comment:
“In general, the physician treats a patient based on diagnosis and condition. After a patient recovers, the physician considers consulting occupational therapists for making a splint. So, I need to know a patient’s condition and prognosis of each case. Sometimes, I have found a broken bone and needed to make an evaluation to select a splint type to fit each case.”

Similarly, Sompong’s experience reflected condition considerations and possible factors that can influence the clinical effectiveness of splinting. This provided the main rationale for splint selection as a part of treatment planning and rehabilitation. As Sompong stated:
“I think that client condition is key to selecting types of splints, and it can help to predict the outcomes after the use of splints. The management of condition needs a specific splint with appropriate duration and time for recovery. Moreover, I need to evaluate current conditions in readiness to use splints. So, I provide splinting for a client through treatment planning and intervention, including maintenance.”

Somyod’s response highlighted the limitations of splinting practice for some patients. There are restrictions on providing splints depending on the patients’ condition and sensory loss. Splinting may pose risks and can affect the bone and skin due to the lack of feeling of pressure, which can lead to damage to the skin. As Somyod stated:
“I found that my patients have sensory problems, especially sensory loss that can affect splint therapy. In those cases, I don’t use splinting for them because the pressure from the splint will affect their bone and skin. It is quite dangerous…it is quite dangerous to hurt them.”

Regarding splint types for different conditions, Somsri indicated that the appropriateness of splinting for the condition is an integral aspect of preventing deformity. The clinical reasoning supporting the occupational therapists’ decisions regarding the choice of splints and the reasons that the therapist uses in practice based on key factors are revealed in their statement:
“We think that [it is] just the general knowledge of health providers that we need to find the relevant and effective interventions to match a patient’s condition. We need to consider the types of splints and raw materials to fit a patient. It will help to fix bone alignment and reduce patient deformities.”


**Theme 2. Effective function and patients’ value**


Most participants expressed the importance of patient function and how what they value can influence the outcomes of their splinting service, which refers to evaluating the clinical effectiveness of mobilization splinting. Participants modified their splints to accommodate patients’ needs throughout the splinting procedure. In Somsak’s account, they reflected on a patient’s need for splinting to reduce spasticity and how it facilitated the capacity of hand function, as stated in the following comment:
“I thoroughly inform my patients regarding what is involved in splint therapy. For example, in one case, a patient wanted to wear an antispastic splint to reduce spasms and pain. He also told me that he needed to wear the splint only at night. In my clinical opinion, I thought that it would not be effective to reduce spasticity and improve hand function when the patient wears a splint and needs to go to sleep. However, I created a splint for him since he actually works at his home during the nighttime.”

The use of splinting practices could either promote or hinder performing activities based on the mechanical properties of the splint design. Sompong’s account indicated that prescribed splint designs recommended by the medical physician did not always meet the client’s needs and applications for daily living. The following comment reflects the effectiveness of splinting with effective function and is also value-added, the value of being able to drive a car, as described in the following comment:
“The orthopedic physician then consulted the occupational therapists to make a static splint to prevent ulnar deviation. When the client wore this kind of splint, he felt good, but he could not bend his fingers. He complained that he could not do anything and that the splint negatively affected his abilities to perform everyday activities. I considered his problem and changed his current splint to a dynamic splint. My client was happy because he could perform activities that he wanted.”

Although there are some problems encountered when providing splints for children, occupational therapists need to motivate pediatric patients to accept these devices. Splinting fabrication may use colored splint material sheets to create a vivid pattern and add cartoon stickers to improve the acceptability of splinting. In Somporn’s interview, she indicated that she motivated pediatric patients to play with toys, and used vivid patterns and stickers in their splint designs. She commented:
“I treat both mature and pediatric patients. The biggest problem is providing splint therapy for children. I have to understand that children are not the same as adults; they have different needs and levels of understanding at different ages. A good way to approach children is to talk to them about their favorite cartoon characters by using stickers that can be stuck on splints or using a vivid splint pattern. Sometimes, I initiated splinting with the child by playing with toys.”


**Theme 3. Knowledge and experiential skills**


Participants reflected on their knowledge and experience of splinting practices and services. Most therapists develop and increase their competence and confidence through everyday splinting application and workshop training including following some form of splint prescription guidelines, which help to promote the therapist’s self-efficacy. This, combined with the body of knowledge related to splinting services, helps to maximize patient outcomes. In Somying’s account, they reflected on the basic knowledge of pathologies, conditions, splinting skills, and clinical experiences including communication within interdisciplinary teams, as stated:
“I believe that we basically need to understand a patient’s condition. This includes understanding cause and effect relationships as well as preventive measures. As occupational therapists, we need to be confident in gathering a patient’s information in order to select the types of splints that fit a patient’s condition. Overall, we create the pattern and deal with a patient to match the affected hand with the appropriate splint. Moreover, we need to check up on splinting mechanisms and their functions and evaluate a patient’s satisfaction including giving information about splint maintenance.”

Increased participation in splinting practice enhances the competence of the splinting service. This is closely related to the knowledge and levels of experience of practitioners that have extensively dealt with different patient conditions, which allows them to fabricate various types of splints that fit their patients’ needs. One participant reflected that occupational therapists must have more practice in splinting design and fitting with their patients to enhance their clinical skills. As Somyod stated:
“At the beginning, I gained experience in splinting design from senior occupational therapists and by servicing patients. When I intervened with the patients more often, I repeated splinting every day. Therefore, I became more familiar with their situation and also more confident about making sure that I can do it better for my patients. Moreover, I actually learned from the lessons that try to draw and design the pattern on paper before fabricating then splinting sheet.”

Somruidee felt that the beginning of her splinting practice was filled with doubt and uncertainty. Integrated knowledge and experiences of splinting support her learning and improve her skills as an occupational therapist when providing a relevant splint to patient-specific conditions. As she commented:
“Fifteen years ago, I was nervous when I served my patients in splinting therapy. When I intervened with patients, I had doubts and uncertainty, and did not know whether there would be good outcomes or not. Nevertheless, I did not quit, and kept going forward to improve myself as a therapist. I opened the door to learn from splinting workshops and repeatedly practiced at my clinic over and over again. Now, I have the experience of how to deal with patients who want to be served with splinting. It takes time, but you can make progress.”

As revealed above, basic knowledge of pathological conditions and splint fabrication is integral for occupational therapists to gain experiential clinical skills with their patients within splinting practice. This understanding underscores occupational therapists’ experience with providing the best possible patient service. Nevertheless, the ongoing learning process and further experience with splinting practice provide the means for occupational therapists to improve their competencies and confidence in the splinting services provided to patients.

## 4. Discussion

In this study, specialist occupational therapy participants provided their perspectives and experiences of several types of splint fabrication used for splinting that are individualized for each patient. These findings provide important contributions to the body of knowledge on splinting practices. Therapists’ reflections on the common use of splinting for the upper limbs and hands in daily service have received relatively little attention within the literature. In this study, we found that the upper limb/hand splints most used by participants in daily service in decreasing order were functional resting, cock-up, thumb spica, and finger splints. The least common splinting practices were supination and weight-bearing splints (30%). Occupational therapists’ preferred splinting in hand function for children with neurological impairment. This finding agrees with those reported by Hepworth et al. [30], who found that functional resting and neoprene thumb abduction splints are recommended in 80% of cases, anti-spasticity splints in 60% of cases, and both serpentine and supination splints in 32.5% of cases.

Previous studies have reflected the relevant perspectives and experiences of occupational therapists working with various specific ages in childhood and areas of neurological conditions. Similarly, we focused on patients with various conditions across their lifespan. However, we determined that the functional resting splint was the preferred therapeutic device, having the highest percentage of use within splinting services. Furthermore, we identified six major conditions (including neuromuscular, orthopedic, degenerative, and developmental conditions) that most Thai occupational therapists face during splinting therapy. However, the literature appears to be limited regarding the proportion of patient conditions treated with splinting services, and it is hard to compare the frequency of patients’ conditions or injuries requiring splinting services within occupational therapy settings.

In terms of conditions and splint types, our findings provide significant information for Thai occupational therapy educators when preparing supporting materials, emphasizing critical factors to boost skills and facilitate experiential learning through workshops and practical experience. The knowledge of the proportion of splint types related to the most common conditions will assist educators to re-arrange both the quantity and quality of time focused on learning outcomes. At this point, the occupational therapy programs in Thailand should be concerned with providing the essential knowledge and practice for entry into the profession. Occupational therapy students should gain knowledge and skill after graduation through on-the-job learning and continuing professional development.

The effective outcomes of evidence-based practice need to be related to the type of approach each patient receives based on individual needs. In this study, participants were asked to consider the aspects of splinting function for their patients. Prevention of bone and joint deformities was ranked first, prevention of muscular contracture was ranked second, and maintaining and increasing range of motion was ranked third when evaluating participants’ perspectives. However, Hepworth et al. [30] evaluated patient factors when providing splinting therapy for children with neurological impairments and reported that maintaining and improving range of motion ranked first (97.5%), preventing contracture was ranked second (92.5%), and compliance with the caregiver ranked third (90%) per the respondents’ perspectives (*n* = 40). In the study of network meta-analysis, hand therapy is important because it proves that Carpometacarpal (CMC) osteoarthritis (OA) patients are best treated with a long thermoplastic carpometacarpal-metacarpophalangeal splint (rigid CMC-MCP) for pain relief, while a short custom-made CMC-MCP splint can be used to improve hand function [31]. In our study, Thumb Spica and De Quervain’s are short-hand splints and showed 80% and 50% in service frequency, and were also used to improve hand function and prehension.

Kilbride et al. [19] explored the contemporary splinting therapy provided by British occupational therapists and physiotherapists for adults with neurological dysfunction. They found that prevention of contractures was ranked first (95%), muscle and joint were ranked second (95%), and alignment and reversal of contractures ranked third (75%) in splinting practice. The previous studies are different from this study due to the differences in culture and occupational therapy training. The prevalence of conditions in the Thai population related to socioeconomics and occupations influences the selection of the type of splints. According to Suphanchaimat et al. [32], Thai economic-growth-related road traffic injuries rose from 449.0 to 524.9 cases per 100,000 population from 2012 to 2016. Economic development occurred throughout the country, causing an increase in road accidents, leading to a high prevalence of nerve injuries and bone fractures, which are common conditions requiring splinting service. This finding should be reflected in the knowledge, skills, and attributes occupational therapy students bring to their placements and future careers.

To accomplish goals in splinting, students need to combine theoretical knowledge and practice. Combined with practical experience, this knowledge leads to increased confidence and competency during the provision of splinting interventions. In our study, most specialist occupational therapists viewed clinical knowledge of the condition as ranking first, splinting skills and experience ranking second, and availability and accessibility ranking third. These factors were the primary considerations for splint selection and application among Thai occupational therapists (Figure 2). Hepworth et al. [30] found that hand splinting in children with a neurological impairment requires knowledge of the condition (92.5%), experience (82.5%), competency (80%), and available resources (75%). Similarly, Chazen and Franzsen [33] evaluated the opinion of experts regarding splinting adult patients with neurological conditions, who reported that occupational therapists should develop skills in fabricating splints and learn from others’ experiences with clinical practice.

As a result of this study’s applicability, it can help bridge the gap between the clinical context and splinting training in an occupational therapy program. Occupational therapy education in splinting and fabrication should emphasize and strengthen the skills and competence of occupational therapy students and practitioners based on the prevalence of the condition and types of splinting practice in everyday service. Nerve injuries and orthopedic cases are the major clients in splinting services. A functional resting splint is the most common splint in practice and service. Implementing splinting service contributes to the prevention of bone and muscle contracture including joint deformities. Individual interviews with occupational therapy specialists should be explored as to how to develop education, and training on proper splint techniques between education (universities) and clinical service (hospitals). A future study would be based on a link between academics and clinical services that contribute to splinting services across Thailand. This will strengthen the splinting course in the occupational therapy curriculum, professional development (propositional, procedural, and personal knowledge), and clinical skills (conditional, procedural, and interactive dimensions).

## 5. Limitations

Limitations of this study included the small sample size due to the inclusion criteria and location context of hand rehabilitation in the city of Bangkok and its outskirts. There are six regions in Thailand, so this study cannot be seen as representative of the whole country. It would be beneficial to study the perspectives of occupational therapists in the six regions throughout Thailand in further studies. Moreover, individual interviews should be extended to all six regions, both rural and urban areas. In addition, this study involved two private hospitals but future research would be extended to more private hospitals to gain a deeper understanding of splinting practices in occupational therapy services across Thailand.

## 6. Conclusions

In summary, a successful splinting development program could be transformed into a more clinically focused professional development opportunity between educators and occupational therapists. In the Thai occupational therapy program, the course on splints and orthoses should provide knowledge and skills in fabricating splints according to patient conditions and relevant splint types, which adds effective function and value for patients in everyday activities. It provides administrators, educators, and practitioners with outcome-based evidence for program revision. The learning outcomes should establish a direct link between the teaching-learning approach (knowledge, skills, and attitude) in universities and clinical services in real circumstances. The further development of the splinting program requires action plans to enable students to become competent practitioners, good clinical reasoners, and life-long learners.

## Figures and Tables

**Figure 1 ijerph-19-08995-f001:**
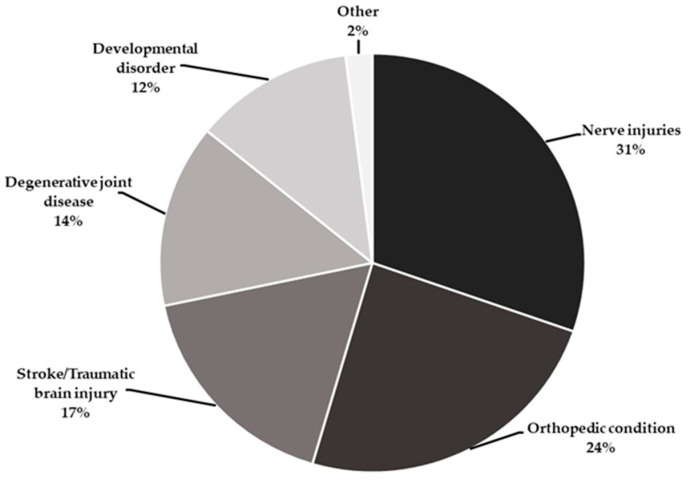
Distribution of clients’ conditions in splinting practice (*n* = 30).

**Figure 2 ijerph-19-08995-f002:**
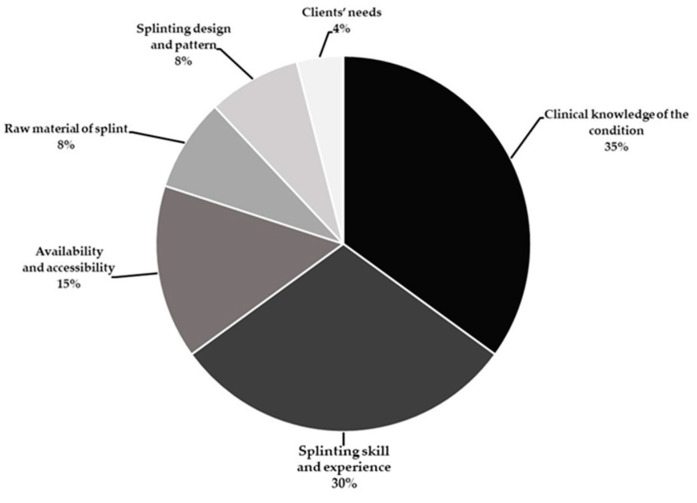
Therapists’ rationale for splinting selection and application (*n* = 30).

**Figure 3 ijerph-19-08995-f003:**
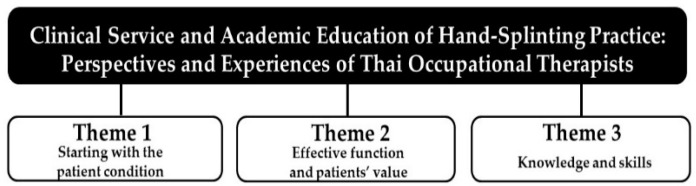
Summary of main themes.

**Table 1 ijerph-19-08995-t001:** Demographic data of the survey participants (*n* = 30).

Workplace/Splinting Practice Experience	2–5 Years	6–10 Years	11–15 Years	16–20 Years	>20 Years	Total
Hospital A	3	-	-	2	1	**6**
Hospital B	-	1	-	1	1	**3**
Hospital C	3	3	1	-	1	**8**
Hospital D	1	1	-	-	1	**3**
Hospital E	2	-	1	1	-	**4**
Hospital F	1	-	1	1	-	**3**
Hospital G	3	-	-	-	-	**3**
**Total**	**13**	**5**	**3**	**5**	**4**	**30**

**Table 2 ijerph-19-08995-t002:** Demographic characteristics of participants in the interview (*n* = 8).

Participant	Sex	Age	Years of Experience in Splinting	Clinical Field	Clinical Setting
Somying	Female	44	18	Physical dysfunction & community	Public hospital
Sompong	Male	40	16	Physical dysfunctions	Private hospital
Somsri	Female	48	24	Physical and pediatrics	Public hospital
Somsak	Male	40	17	Physical and pediatrics	Public hospital
Somchai	Male	47	24	Physical dysfunction	Public hospital
Somyod	Male	49	25	Physical dysfunction	Private hospital
Somruidee	Female	38	15	Physical and elderly	Public hospital
Sompon	Female	45	21	Physical dysfunction	Public hospital

**Table 3 ijerph-19-08995-t003:** Service frequency of various splint types (*n* = 30).

Type of Splint	Used (n)	Used (%)	Never Used (n)	Never Used (%)
Functional Resting *	30	100	0	0
Cock-up **	28	93.3	2	6.7
Thumb spica ***	24	80	6	20
Finger ***	24	80	6	20
De Quervain’s	15	50	15	15
Anti-spastic	14	46.6	16	53.4
Dynamic	9	30	21	70
Supination	1	3.3	29	96.7
Weight-bearing	1	3.3	29	96.7

Note. Ranked * first, ** second, and *** third in splint use.

**Table 4 ijerph-19-08995-t004:** Client factors in splinting practice (*n* = 30).

Client Factors in Splinting Practice	Very Frequently (%)	Frequently (%)	Occasionally (%)	Rarely (%)	Never (%)
Prevention of bone and joint deformities	43	53	4	-	-
Prevention of muscle contracture	40	46	7	7	-
Maintaining/increasing range of motion	33	53	14	-	-
Reducing muscle tone	27	50	13	3	7
Improving hand function and prehension	10	47	20	17	6
Patients’ needs	7	13	20	47	13
Pain relief	6	13	41	27	13

## Data Availability

Data are available from the corresponding author upon reasonable request.

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
