# Peer review of "Bridging the Gap between Clinical Service and Academic Education of Hand-Splinting Practice: Perspectives and Experiences of Thai Occupational Therapists"

_ijerph, 2022, doi:10.3390/ijerph19158995_

Round 1

Reviewer 1 Report

Thank you for the opportunity to review this study exploring Perspectives and Experiences of Occupational Therapists in the Thai Context. At this point in time, this study has some limitations and a considerable amount of work needs to be done to improve its quality. In my humble opinion, the article is very complex to understand. Authors need to review the presentation of their idea. My main areas of concerns are described below:

-       Add more information about clinical reasoning and education process.

-       Describe more information about Thailand education. How long does occupational therapy training take?

-       Please, synthesize the aim of your study.

-       Why did you decide as inclusion criteria having more than 10 years experiences in splinting service?

-       This study was carried out in Bangkok and the surrounding areas. Then, title should be change.

-       Please clarify. According to inclusion criteria, only OT with more that 10 years experiences would be include. Why do you have OT with 2-5 years experiences? It is very hard and difficult to understand the methodology carried out in this study and the results obtained.

-       In my opinion, 30 participants are not representative of the Tai context. Please, add a limitation section.

Author Response

Response to Reviewer 1 Comments

We greatly appreciate your thoughtful comments concerning our manuscript entitled " Bridging the Gap between Clinical Service and Academic Education of Hand-Splinting Practice: Perspectives and Experiences of Occupational Therapists in the Thai Context" (Manuscript ID: ijerph-1766887). Those comments are all valuable and very helpful for revising and improving our paper. We have studied the comments carefully and have made correction which we hope to meet your approval. Revised portions are marked in red highlighting in the paper.

The main corrections in the paper and the responses to the reviewer’s comments are as flow:

Point 1: Add more information about clinical reasoning and education process.

Response 1: We are thankful and agreed with your comments and suggestion about the clinical reasoning and education process on the page 1 in line 63-76.

A major goal of undergraduate occupational therapy programs is to prepare competent and capable entry-level therapist (Koski, Simon & Dooley, 2013). Clinical reasoning skills are one of the most challenging aspects of the occupational therapy program (Dancza et al., 2013). As a general term, clinical reasoning can be described as the process of gathering and interpreting information in order to make decisions regarding health care (Tiffen et al, 2014).  Other terms used interchangeably with clinical reasoning are, clinical judgment, decision-making process, and critical thinking reasoning (Simmons, 2010). In health professional education, clinical reasoning is considered a core component of teaching and assessment (Young et al, 2018a). A progressive approach to clinical reasoning skills should be employed during the education program, according to the European Network of Occupational Therapists (ENOTHE, 2004). Occupational therapists should supervise final-year students during their fieldwork education to ensure learning takes place at a deeper level of reasoning (Hills, Ryan, Smith & War-ren-Forward, 2012). Hence, clinical reasoning is a primary skill in OT education for identifying, prioritizing, establishing plans, and interpreting clinical data, which are all important to occupational therapy practice.

Point 2: Describe more information about Thailand education. How long does occupational therapy training take?.

Response 2: We are pleased that you suggested more information about Thailand education. The manuscript revised version has more information about Thailand education and the duration for studying occupational therapy on page 2 in the Lines 52-62. 

The Ministry of Higher Education, Science, Research, and Innovation enacted laws and regulations for Thai universities. They require that Thai Qualifications Framework (TQF) must be implemented for Higher Education (Office of the Higher Education commission, 2009) and the Professional Standards for the Art of Healing in OT (Ministry of Public Health, 2011). Students take 4 years to earn their Bachelor of Science (Occupational Therapy) with an eight semester curriculum. In the beginning of OT education, OT students understood time of study, which they must engage in course journey to achieve the learning outcomes. The Thai OT program is comprised of general education courses, specification or professional courses, and free elective courses. Thailand meets the World Federation of Occupational Therapists (WFOT) requirement that OT students complete a minimum of 1,000 hours of fieldwork experience as part of the OT curriculum.

Point 3: Please, synthesize the aim of your study.

Response 3: We are thankful and agreed with your comment and suggestion. We have synthesized the aim of our study on pages 3-4 in line 148-160.

The purpose of this mixed methods sequential explanatory study was to identify factors contributing to occupational therapists’ perspectives in splinting practice. We did this by obtaining results from a survey of 42 occupational therapists and then following up with eight purposefully selected individuals. The eight individuals were selected to explore their experience in splining practice in more depth through a thematic analysis. The specific objectives were:

  • To explore occupational therapists’ perspective in clients’ conditions and factors, splint types and rationale for splinting selection and practice.
  • To find the experience of expert occupational therapists in the nature of clinical instructors determining their clinical reasoning ability and skill in splinting practice.

The outcomes of the two specific objectives will help shape the scope and content of courses on splinting and orthotics in occupational therapy programme revisions.

Point 4: Why did you decide as inclusion criteria having more than 10 years experiences in splinting service?

Response 4: Thank you for the comment. For the inclusion criteria, we are looking for occupational therapists with significant experiences in splinting practice. This suggestion has been corrected on page 5 in line 219-220, in the section of interview participants and recruitment.

(2) having more than 10 years of experience and identify as an expert in splinting service.; (3) working as a clinical instructor.

Point 5: This study was carried out in Bangkok and the surrounding areas. Then, title should be change.

Response 5: We would like to thank you for your mention about the title related to the capital city and suburb areas. We already made it to be clear about the title: “Bridging the Gap between Clinical Service and Academic Education of Hand-Splinting Practice: Perspectives and Experiences of Thai Occupational Therapists”.

Point 6: Please clarify. According to inclusion criteria, only OT with more that 10 years experiences would be include. Why do you have OT with 2-5 years experiences? It is very hard and difficult to understand the methodology carried out in this study and the results obtained.

Response 6: We thank the reviewer for the comments and recommendations. This study is mixed methods design that integrated quantitative and qualitative studies.

First, the quantitative survey study included participants who are occupational therapist with 2 years until more than 20 years of experience.

While the second is a qualitative study. For this portion, we are looking for occupational therapist who are working as a clinical instructor (CI), have more than 10 years of experience and identify as an expert in splinting service. For these individuals we did an in-depth interview and learned from their expertise with clinical experience.

Point 7: In my opinion, 30 participants are not representative of the Tai context. Please, add a limitation section.

Response 7: We truly appreciated your comments about 30 participants are not representative of the Thai context. Now we addressed it into the limitation section on the page 14 in line 559-562.

This present study was the first survey in splinting from Thai occupational therapists and also presented experience in therapy service. Limitations of this study included that the participants of the survey showed a small sample size due to inclusion criteria and location context of hand rehabilitation in the city of Bangkok and outskirt areas.

Special thanks to you for your good comments. We tried our best to improve the manuscript and made some changes in the manuscript.

We appreciate the Editors/Reviewers’ warm work and hope that the correction will meet your approval. Once again, thank you very much for your comments and suggestions.

Best regards,

Anuchart Kaunnil and research team

Reviewer 2 Report

Thank you the authors for considering the present journal to publish their investigation. It's a interesting topic that concerns the differences between hand splinting from education to practice. As occupational therapy, I totally understand the problematic. However, I think the authors must improve methodological aspects of the study.

1. Please explain what type of mixed methods you use, where is quantitative and where is qualitative, and also explain the tools you used in the different methodologies. It seems to be a qualitative study, not a mixed-method study.  So, review the literature about mixed methods and explain it in detail. 

2. In the discussion section please explain the applicability of the present study, what it adds to the existing literature.

The topic is interesting but it's necessary that the methodology was appropriated, also I think if the authors continue telling it's a mixed methods study, must rewrite the objetive.

Author Response

Response to Reviewer 2 Comments

We greatly appreciate your thoughtful comments concerning our manuscript entitled " Bridging the Gap between Clinical Service and Academic Education of Hand-Splinting Practice: Perspectives and Experiences of Thai Occupational Therapists" (Manuscript ID: ijerph-1766887). Those comments are all valuable and very helpful for revising and improving our paper. We have studied the comments carefully and have made correction which we hope to meet your approval. Revised portions are marked in red highlighting in the paper.

The main corrections in the paper and the responds to the reviewer’s comments are as flows:

Point 1: Please explain what type of mixed methods you use, where is quantitative and where is qualitative, and also explain the tools you used in the different methodologies. It seems to be a qualitative study, not a mixed-method study. So, review the literature about mixed methods and explain it in detail.

Response 1: We are thankful and agreed with your comments and suggestion about the type of mixed methods and tools to collect data. The manuscript revise version showed this point in the abstract section on page 1 line 21-29 to the to be clear in reading for the reader.

An explanatory sequential mixed methods design was used, and it involved collecting quantitative data first and then explaining the quantitative results with in-depth qualitative data. In the first quantitative phase of the study, a questionnaire survey was mailed to occupational therapists. The questions were regarding contemporary hand splinting practices in clinical use at seven hospitals in the capital city of Bangkok and outskirt areas. This was done to explore their perspective on splinting practice. The second phase was qualitative and conducted as a follow-up to the quantitative results to support the quantitative results. Semi-structured interview were completed to investigate occupational therapists who were experts in splinting practice at the same hospital settings. These transcripts were analyzed using thematic analysis.

Now we addressed about the type of mixed methods and tools into the Materials and Methods section on the page 13 in line 162-165.

This study addressed the perspectives and experiences of occupational therapists on the upper limb and hand splinting practices. An explanatory sequential mixed methods design [23] (Creswell & Plano Clark, 2018) was used, and it involved collecting quantitative data first and then explaining the quantitative results with in-depth qualitative data.

Point 2: In the discussion section please explain the applicability of the present study, what it adds to the existing literature.

Response 2: We are pleased that you suggested more the information about the applicability of this present study. The revised manuscript shows this point in the discussion section on the page 13-14 in the Line 541-556.

As a result of this study's applicability, it can help bridge the gap between clinical circumstances and splinting training in an occupational therapy programme. Occupational therapy education in splinting and fabrication should emphasize and strengthen the skills and competence of occupational therapy students and practitioners based on the prevalence of the condition and types of splinting practice in everyday service. Nerve injures and orthopedic cases are the major clients in splinting services. Functional resting splint is the most common splint in practice and service. Implementing splinting service contributes to prevention of bone and muscle contracture including joint deformities. Future studies should be surveyed perspective of occupational therapists from the six regions of Thailand. Individual interview with occupational therapy specialist should be explored how to develop education, training on proper splint technique between the parts of education (universities) and clinical service (hospitals). The future study would be a link between academic and service that contributed to splinting services across Thailand. This will strengthen splinting course in the occupational therapy curriculum, professional development (propositional, procedural, and person-al knowledge) and clinical skills (conditional, procedural, and interactive dimensions).

Point 3: The topic is interesting but it's necessary that the methodology was appropriated, also I think if the authors continue telling it's a mixed methods study, must rewrite the objective.

Response 3: We thank the reviewer for this useful comment and consideration that it is a good to rewrite the objective. So, we addressed and elaborated in text for better understanding among the readers in the introduction section on page 3 in line 148-153.

The purpose of this mixed methods sequential explanatory study was to identify fac-tors contributing to occupational therapists’ perspectives in splinting practice. We did this by obtaining results from a survey of 42 occupational therapists and then following up with eight purposefully selected individuals. The eight individuals were select-ed to explore their experience in splining practice in more depth through a thematic analysis.

Special thanks to you for your good comments and suggestion. We tried our best to improve the manuscript and made some changes in the manuscript.

We appreciate the Editors/Reviewers’ warm work and hope that the correction will meet your approval. Once again, thank you very much for your comments and suggestions.

Best regards,

Anuchart Kaunnil and research team

Reviewer 3 Report

Manuscript ID: ijerph-1766887

Manuscript title: Bridging the Gap between Clinical Service and Academic Education of Hand-Splinting Practice: Perspectives and Experiences of Occupational Therapists in the Thai Context

Comments

This manuscript reports a mixed-method (both survey data and interview) analysis of the perspectives and experiences of occupational therapists on the upper limb and hand splinting practices in Bangkok and outskirt. The Introduction is well written, with a clear background and rationale for this study. The methods are adequate to the proposed aims and are described in sufficient details to maximize reproducibility. Particularly, the authors reported high standards for data handling, thematic analysis, rigor, and trustworthiness that increases the confidence in the findings. The conclusions seem supported by the presented data. I have only a few comments for the authors to consider.

Major comments

1. Discussion. It would be interesting to discuss whether the most common use of splinting for the upper limbs and hands are supported by current evidence on the topic. Unsurprisingly, it may be the case that most common practices are not necessarily backed up by high-quality (low biased) evidence.

Minor comments

1. As in Figure 2, the horizontal bars in Figure 1 could be presented in decreasing (or increasing) order of % values for a clearer interpretation.

2. Likewise, Tables 3 and 4 could be presented in decreasing (or increasing) order of use for a clearer interpretation.

Author Response

Response to Reviewer 3 Comments

We greatly appreciate your thoughtful comments concerning our manuscript entitled " Bridging the Gap between Clinical Service and Academic Edu-cation of Hand-Splinting Practice: Perspectives and Experiences of Thai Occupational Therapists" (Manuscript ID: ijerph-1766887). Those comments are all valuable and very helpful for revising and improving our paper. We have studied the comments carefully and have made correction which we hope meet your approval. Revised portion are marked in red highlighting in the paper.

The main corrections in the paper and the responds to the reviewer’s comments are as flows:

Point 1: Discussion. It would be interesting to discuss whether the most common use of splinting for the upper limbs and hands are supported by current evidence on the topic. Unsurprisingly, it may be the case that most common practices are not necessarily backed up by high-quality (low biased) evidence.

Response 1: We are thankful and agreed with your comments and suggestion about common splints not backed by evidnece. So, we revised and deleted this evidence on page 12 in the discussion section.

We deleted the paragraph about Reid surveyed 174 occupational therapists in the use of hand splints for children with neuromuscular dysfunction, finding that resting palmar splints (71.1%), thumb (65.4%), and the spastic reduction splints (32.9%) were used.

Point 2: As in Figure 2, the horizontal bars in Figure 1 could be presented in decreasing (or increasing) order of % values for a clearer interpretation.

Response 2: We truly appreciated you mentioned about the Figure 2 and the horizontal bars in Figure 1. We elaborated and made it to be more clear the Figure 1 on page 7 and Figure 2 on page 9 for better understanding among the readers.

Point 3: Likewise, Tables 3 and 4 could be presented in decreasing (or increasing) order of use for a clearer interpretation.

Response 3: We agree with your comments and addressed it to allow for a clearer interpretation in the Table 3 on page 7 and Table 4 on page 8.

Special thanks to you for your good comments and suggestion. We tried our best to improve the manuscript and made some changes in the manuscript.

We appreciate the Editors/Reviewers’ warm work and hope that the correction will meet your approval. Once again, thank you very much for your comments and suggestions.

Best regards,

Anuchart Kaunnil and research team

Reviewer 4 Report

Dear Authors,

Thanks for the opportunity to review, I must say that qualitative studies generally lack methodology, but you have provided solid methodological points, but I must suggest:

L22 superfluous

In the abstract methods, describe the type of target participant and list the settings achieved, rather than the number of participants and that's it.

Bibliographic references regarding splinting are a bit dated I suggest to amplify the type of splint that can be used and reachable by the patient cohort for example .. ref: https://doi.org/10.1016/j.apmr.2020.06.012

L128 who participated in the construction of the questionnaire? Do you have references in literature?

L140 reference missing

I suggest a figure to accompany the thematic analysis, to increase the perspicacity of the work and help the reader to read.

Author Response

Response to Reviewer 4 Comments

We greatly appreciate your thoughtful comments concerning our manuscript entitled " Bridging the Gap between Clinical Service and Academic Edu-cation of Hand-Splinting Practice: Perspectives and Experiences of Thai Occupational Therapists" (Manuscript ID: ijerph-1766887). Those comments are all valuable and very helpful for revising and improving our paper. We have studied the comments carefully and have made correction which we hope meet your approval. Revised portion are marked in red highlighting in the paper.

The main corrections in the paper and the responds to the reviewer’s comments are as flows:

Point 1: Point 1: Thanks for the opportunity to review, I must say that qualitative studies generally lack methodology, but you have provided solid methodological points, but I must suggest:

Response 1: We are thankful and agreed with your comments and suggestion about the type of mixed methods and tools to collect data. We addressed the type of mixed methods and tools into the Materials and Methods section on the page 4 in line 162-165.

This study addressed the perspectives and experiences of occupational therapists on the upper limb and hand splinting practices. An explanatory sequential mixed methods design [23] (Creswell & Plano Clark, 2018) was used, and it involved collecting quantitative data first and then explaining the quantitative results with in-depth qualitative data.

Point 2: L22 superfluous, In the abstract methods, describe the type of target participant and list the settings achieved, rather than the number of participants and that's it.

Response 2: We are thankful and agreed with your comments and suggestion about the type of target participant and list the setting achieved on the page 1 in line 21-29.

An explanatory sequential mixed methods design was used, and it involved collecting quantitative data first and then explaining the quantitative results with in-depth qualitative data. In the first quantitative phase of the study, a questionnaire survey was mailed to occupational therapists. The questions were regarding contemporary hand splinting practices in clinical use at seven hospitals in the capital city of Bangkok and outskirt areas. This was done to explore their perspective on splinting practice. The second phase was qualitative and conducted as a follow-up to the quantitative results to support the quantitative results. Semi-structured interview were completed to investigate occupational therapists who were experts in splinting practice at the same hospital set-tings. These transcripts were analyzed using thematic analysis.

Point 3: Bibliographic references regarding splinting are a bit dated I suggest to amplify the type of splint that can be used and reachable by the patient cohort for example .. ref: https://doi.org/10.1016/j.apmr.2020.06.012

Response 3: We thank the reviewer for this useful suggestion about this systematic review and network meta-analysis. Now we addressed it into the discussion section on the page 13 in line 507-513.

In the study of network meta-analysis, hand therapy is important because it proves that Carpometacarpal (CMC) osteoarthritis (OA) patients are best treated with a long thermoplastic carpometacarpal-metacarpophalangeal splint (rigid CMC-MCP) splint for pain relief, while a short custom-made CMC-MCP splint can be used to improve hand function [30] (Marotta et al., 2021). In our study, Thumb Spica and De Quavain’s are a short hand splint and showed 80% and 50% in service frequency, and also used to improve hand function and prehension.

Point 4: L128 who participated in the construction of the questionnaire? Do you have references in literature?

Response 4: We truly appreciated your comments about the construction of the questionnaire. We addressed information to clarify who participated in the construction of the questionnaire survey of 2.1. Survey Procedure on on page 4 in line 169-172.

A questionnaire survey was developed by the research team and contained questions in five parts: 1) demographic profile, workplace location, and occupational therapy specialty; 2) experience in splinting production; 3) types of splinting designs used in practice and fabrication; 4) patient factors for splinting practice; 5) therapist factors in splinting practice.

And validated by three experts on page 4 in line 178-180.

The questionnaire was validated by three external occupational therapists who had more than 20 years’ experience with splinting. This questionnaire received a score of 0.73 on the Item Objective Congruence (IOC) index.

Point 5: L140 reference missing

Response 5: We would like to apologize for this mistake and thank you for your comments and suggestion. We have now improved the reference and now it is on the page 4 line 173-177 and in the reference number 24.

Prior to administering the survey, Item Objective Congruence (IOC) was used to evaluate the items of the questionnaire based on the scoring system from –1 to +1 (congruent/agree = +1, questionable = 0, and incongruent/disagree = -1). The questionnaire items with scores higher than or equal to 0.5 were retained. Conversely, the items with scores lower than 0.5 were revised [24].

  1. Rovinelli, R.J.; Hambleton, R.K. On the use of content specialists in the assessment of criterion-referenced test item validity Dutch Journal of Educational Research 1977, 2, 49-60.

Point 6: I suggest a figure to accompany the thematic analysis, to increase the perspicacity of the work and help the reader to read.

Response 6: Thank you very indeed for the comment, we added the figure to accompany to help the reader. Now we addressed it into the discussion section on the page 9 in line 335-337.

Special thanks to you for your good comments and suggestion. We tried our best to improve the manuscript and made some changes in the manuscript.

We appreciate the Editors/Reviewers’ hard work and hope that the correction will meet your approval. Once again, thank you very much for your comments and suggestions.

Best regards,

Anuchart Kaunnil and research team

Round 2

Reviewer 2 Report

The authors clarified my concerns. Congratulations on your work. It's an interesting paper in the occupational therapy literature. 

Author Response

Response to Reviewer 2 Comments

We thank an anonymous reviewer for your constructive comments. We greatly appreciate your thoughtful comments concerning our manuscript entitled "Bridging the Gap between Clinical Service and Academic Education of Hand-Splinting Practice: Perspectives and Experiences of Thai Occupational Therapists" (Manuscript ID: ijerph-1766887). We are able to respond to the reviewer’s comments, and in doing produce a much-improved manuscript.

Point 1: The authors clarified my concerns. Congratulations on your work. It's an interesting paper in the occupational therapy literature.

Response 1: We are thankful and agreed with your comments and suggestion to make our manuscript revision to be clear in reading for the reader.

We appreciate the anonymous reviewers’ warm work and hope that the correction will meet your approval. Once again, thank you very much indeed for your comments and suggestions.

Best regards,

Anuchart Kaunnil and research team

Reviewer 4 Report

Dear Authors, Thanks for your efforts in revising your manuscript, especially regarding a qualitative analysis, I have only minor concerns:

Line 51 and line 107 are not required as they are conventionally implied in an introduction to provide a rationale for the manuscript.

For percentage values I would suggest pie charts

I would expand the section on limitations, with regard to the methodology of the qualitative analysis itself. Furthermore, the first sentence seems more of a strenght than a limitation.

Author Response

Response to Reviewer 4 Comments

We thank an anonymous reviewer for your constructive comments. We greatly appreciate your thoughtful comments concerning our manuscript entitled "Bridging the Gap between Clinical Service and Academic Education of Hand-Splinting Practice: Perspectives and Experiences of Thai Occupational Therapists" (Manuscript ID: ijerph-1766887). The comments are all valuable and very helpful for revising and improving our paper. We are able to respond to the reviewer’s comments, and in doing produce a much-improved manuscript.

The main corrections in the paper and the responds to the reviewer’s comments are as flows:

Dear Authors, Thanks for your efforts in revising your manuscript, especially regarding a qualitative analysis, I have only minor concerns:

Point 1: Line 51 and line 107 are not required as they are conventionally implied in an introduction to provide a rationale for the manuscript.

Response 1: We are thankful and agreed with your comments and suggestion. We already deleted the conventional patterns about “Historical background” and “Literature review”. Your recommendation is useful for the revised version.

Point 2: For percentage values I would suggest pie charts

Response 2: We truly appreciated your suggestion. Now we addressed the pie charts for the percentage value on page 7 in lines 276-277, and page 8 in lines 306-307.

Point 3: I would expand the section on limitations, with regard to the methodology of the qualitative analysis itself. Furthermore, the first sentence seems more of a strength than a limitation.

Response 3: We would like to apologize for this mistake and thank you for your comments and suggestion. We deleted the first sentence. Your recommendation is useful for the revised version. We had expanded the section on the limitation on pages 13-14 in lines 543-551.

Limitations of this study included the small sample size due to the inclusion criteria and location context of hand rehabilitation in the city of Bangkok and outskirt areas. There are six regions in Thailand, this study cannot be seen as a representative of the whole country. It would be beneficial to study the perspectives of occupational therapists in the six regions throughout Thailand in future studies. Moreover, individual interviews should be extended to all six regions, both rural and urban areas. In addition, this study involved two private hospitals, which in future research would be explored to more private hospitals to gain a deeper understanding of how splinting practices are related to client conditions, splint types, and experiential skills in occupational therapy services across Thailand.

We appreciate the anonymous reviewers’ warm work and hope that the correction will meet your approval. Once again, thank you very much indeed for your comments and suggestions.

Best regards,

Anuchart Kaunnil and research team
